# FreqMamba: Viewing Mamba from a Frequency Perspective for Image Deraining

## ABSTRACT

Images corrupted by rain streaks often lose vital frequency information for perception, and image deraining aims to solve this issue which relies on global and local degradation modeling. Recent studies have witnessed the effectiveness and efficiency of Mamba for perceiving global and local information based on its exploiting local correlation among patches, however, rarely attempts have been explored to extend it with frequency analysis for image deraining, limiting its ability to perceive global degradation that is relevant to frequency modeling (e.g. Fourier transform). In this paper, we propose FreqMamba, an effective and efficient paradigm that leverages the complementary between Mamba and frequency analysis for image deraining. The core of our method lies in extending Mamba with frequency analysis from two perspectives: extending it with frequency-band for exploiting frequency correlation, and connecting it with Fourier transform for global degradation modeling. Specifically, FreqMamba introduces complementary triple interaction structures including spatial Mamba, frequency band Mamba, and Fourier global modeling. Frequency band Mamba decomposes the image into sub-bands of different frequencies to allow 2D scanning from the frequency dimension. Furthermore, leveraging Mamba's unique data-dependent properties, we use rainy images at different scales to provide degradation priors to the network, thereby facilitating efficient training. Extensive experiments show that our method outperforms state-of-the-art methods both visually and quantitatively.

## CCS CONCEPTS

• **Computing methodologies → Image manipulation**.

## KEYWORDS

Image deraining, Frequency analysis, State space model

## 1 INTRODUCTION

Images taken in rainy conditions suffer from significant quality degradation in terms of object details and contrast caused by raindrops in the air, which results in unpleasant visual results and loss of frequency information. Such degradation has a serious adverse impact on the performance of advanced visual tasks such as image classification and target detection. Therefore, image deraining is an important task in the low-level vision field. However, recovering clear images from those with severe raindrop degradation is pretty a

**Unpublished working draft. Not for distribution.**

**Figure 1: Comparison of different modeling methods. Our FreqMamba enhances Mamba's 2D global perception capability from the frequency perspective. Meanwhile, Mamba modeling in frequency dimension is introduced to realize the seamless transition between the spatial and frequency domains.**

difficult job due to the complex coupling of raindrops to the background and the loss of important perception frequency information.

Traditional methods have relied on prior knowledge to dissect the physical properties of rain and background layers, introducing various methods to distinguish rain streaks from clean images. However, these priors, being based on specific observations, might not be reliable for modeling intrinsic features of images or estimating transmission maps in physical models. The advent of deep learning has heralded new directions in rain removal techniques. Many methods are proposed to improve the performance of rain removal methods from different perspectives[8, 14, 35, 44]. Especially, effective global degradation modeling has proved crucial for addressing intricate challenges of image deraining. For instance, the attention mechanism in Transformer achieves great success with the capacity to model image-internal correlations. While, attention mechanisms face scalability challenges due to their quadratic complexity, posing a significant challenge when addressing large images.

Recently, an improved structured state-space sequence model (S4) with a selective scanning mechanism, Mamba, stands out due to its ability to model long-range sequence relationships with linear complexity. Specifically, Mamba's selective methodology can explicitly build the correlation among image patches, further enabling the guidance of clean to degraded areas. However, it is noteworthy that Mamba's approach to processing visual tasks is fundamentally based on pixel sequences. While this allows it to model long-range dependencies, the selective modeling of one-dimensional sequences limits its ability in "global degradation perception" like the Fourier transform. Recognizing this, we attempt to extend it from the perspective of frequency analysis for image deraining.

We integrate two typical frequency analysis techniques into our method: Fourier transform (FT) and wavelet packet transform (WPT).

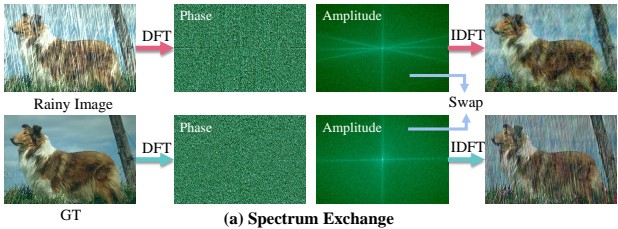

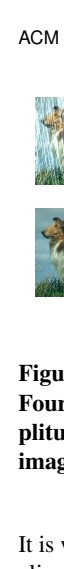

(a) Spectrum Exchange

**Figure 2: Observation of the spectrum exchange of the Discrete Fourier Transform(DFT). The degradation is mainly in the amplitude component, and the Fourier transform can disentangle image content and degradation to some extent.**

It is well known that the Fourier transform has effective global modeling capabilities. In addition, it has good degradation separation properties (Fig. 2). This unique perspective is invaluable for tasks such as image deraining, where understanding the entire image structure can significantly enhance the quality of the restoration result. However, there are inherent gaps between different domains, so simply combining the Fourier transform with Mamba modeling can be quite blunt. We need an intermediate state for a seamless transition. The wavelet packet transform decomposes the image into a series of sub-bands with varying frequencies in the spatial domain, encapsulating both frequency analysis and spatial information simultaneously. By utilizing the wavelet domain as the intermediate state between the spatial and Fourier domains, we establish a smoother transition, enhancing the overall effectiveness of the analysis.

Recognizing the potential for perceiving global degradation via Frequency analysis and Mamba's ability to capture regional correlations within the spatial domain, we introduce FreqMamba. This effective and efficient paradigm utilizes the complementarity between Mamba and frequency analysis for image deraining. The cornerstone of our approach lies in a three-branch structure.

- **Spatial Mamba**: This branch operates on the original image features. It extracts details and correlations within the image, providing crucial insights into degradation patterns.
- **Frequency Band Mamba**: This branch employs the WPT to dissect the input features into a spectrum of features spread across various frequency bands. Arranging these low-resolution features back to the original scale in frequency order, we perform a frequency dimension scanning from low to high frequency as well as in reverse as shown in Fig. 3. This strategy not only enriches the model's analytical breadth but also acts as a pivotal bridge, melding the spatial and Fourier domains into a unified analysis framework, offering a novel perspective on modeling.
- **Fourier Modeling**: This branch equips the model with the ability to conduct global analysis by leveraging the Fourier Transform to process the input. It captures the overarching degradation patterns that affect the image, offering a panoramic view of the frequency spectrum. This global modeling capability is instrumental in comprehensively understanding and mitigating the effects of degradation, ensuring an in-depth removal of rain streaks from the image for a cleaner visual outcome.

Together, these branches form a robust architecture for tackling the challenge of image deraining. The Spatial Scanning Mamba analyzes intricate spatial details, while the Fourier Modeling branch offers a holistic view, enabling the model to understand global degradation phenomena. The Frequency Band Scanning Mamba explores the frequency dimension and provides a new perspective for 2D modeling. Furthermore, leveraging the unique data-dependent characteristics of Mamba, we apply this methodology to rainy images across different scales to derive attention maps based on degradation prior. These attention maps are then integrated into the backbone network to assist in effective training.

## 2 RELATED WORK

### 2.1 Image Deraining

Image deraining has witnessed substantial evolution, transitioning from early model-based strategies to advanced data-driven techniques. Initially, model-driven methods[20, 27], separate rain streaks from the background using hand-crafted features and physical priors. These approaches, while insightful, often struggle with complex rain patterns and diverse real-world scenarios, leading to limitations in their practical applicability and performance.

The advent of deep learning ushered in a new era for image deraining, with data-driven approaches demonstrating remarkable adeptness in extracting and learning features directly from data. The introduction of CNNs marked a significant advancement, enabling more nuanced and adaptive handling of rain streaks across a wide array of images [7, 47, 53]. Besides, the development of architectures incorporating attention mechanisms [38, 40, 44] has further refined the capacity to discern and eliminate rain components, addressing previous shortcomings in model generalization and detail preservation. In this work, we propose a novel baseline with a novel triple interaction block to improve the deraining performance jointly.

### 2.2 Frequency Analysis

The Fourier Transform is a fundamental technique in frequency domain analysis, enabling the conversion of signals to a domain where their global statistical properties can be more easily analyzed. This capability is extensively leveraged across various computer vision tasks. By adept modeling of global domain information, the FT facilitates advancements in various areas [17, 22, 23, 32, 45, 48, 57]. In terms of image restoration, FECNet[16] highlights the utility of Fourier feature amplitudes in isolating global lightness components, thus improving the aesthetic appeal and clarity of images. Similarly, FSDGN [50] reveals how the amplitude of Fourier features serves as a key indicator of global haze information in image dehazing tasks. Despite these advances, the inherent constraints of FT in signal processing hint at untapped potential, suggesting that the efficacy of these methodologies could be augmented further.

Beyond Fourier transform, the Wavelet Transform (WT) is also a mathematical tool used in the analysis of signals and images, offering a complementary perspective to FT. Unlike FT, which excels in capturing frequency information, Wavelet transform provides a multi-resolution analysis that is particularly effective in detecting and representing localized variations in signals.

## 2.3 State Space Models

State Space Models (SSMs)[11, 37] have garnered significant attention in recent years for their ability to effectively model long-range dependencies while exhibiting linear scalability with sequence length. The foundational work of S4 introduced by [11] laid the groundwork for deep state-space modeling, demonstrating promising linear scaling properties. A recent innovation, [31] enhances SSMs' capabilities by integrating gating units. Furthermore, Mamba [10], a data-dependent SSM featuring a selective mechanism and efficient hardware design, has emerged as a standout performer, surpassing Transformers in natural language tasks while maintaining linear scalability with input length. The applicability of SSMs extends beyond NLP, with pioneering works leveraging Mamba for various vision tasks, including image classification [26], biomedical image segmentation [9, 29], and others [3, 34, 39].

## 3 METHOD

In this section, we first introduce the motivation of our proposed method, then introduce the preliminaries of frequency analysis and SSMs. Finally, we outline the overall framework of FreqMamba.

### 3.1 Motivation

Degraded images, particularly those afflicted by rain, suffer from both global degradation and local detail destruction. Frequency-based methods are utilized for mitigating global degradation, leveraging the significant correlation between raindrop effects and the Fourier amplitude spectrum (refer to Fig. 2). Nonetheless, due to the inherent gap between the frequency and spatial domain, the global nature of frequency domain operations makes it impossible to model local dependencies of the spatial domain. Mamba is a state space model distinguished by its selective scanning mechanism, which skillfully facilitates the explicit modeling of interactions among sequences with linear complexity. Applied to vision tasks, it can effectively establish correlations between areas in 2D images.

Recognizing the complementary strengths of frequency-based methods and the Mamba model in addressing different aspects of image degradation, we introduce the FreqSSM Block. This novel structure features a three-branch design engineered to transition smoothly from the correction of global degradation to the refinement of local content. Further leveraging Mamba's distinctive data-dependent properties, we employ degraded images across various scales to derive attention maps and add them in the encoder stage, thereby enhancing the efficiency of training. Expanding upon these foundations, we propose FreqMamba, a specialized solution designed specifically to solve the single-image deraining task.

### 3.2 Preliminaries

*3.2.1 Frequency Analysis in Digital Imaging.* The Fourier Transform is a mathematical technique for transforming a signal from its original domain (often time or space) into a representation in the frequency domain and vice versa with the inverse Fourier Transform (iFT). Specifically for a single-channel image $x$ with size of $H \times W$, the Discrete Fourier Transform is defined as:

$$\mathcal{F}(x)(u,v) = \sum_{h=0}^{H-1} \sum_{w=0}^{W-1} x(h,w) e^{-j2\pi(\frac{h}{H}u + \frac{w}{W}v)}. \quad (1)$$

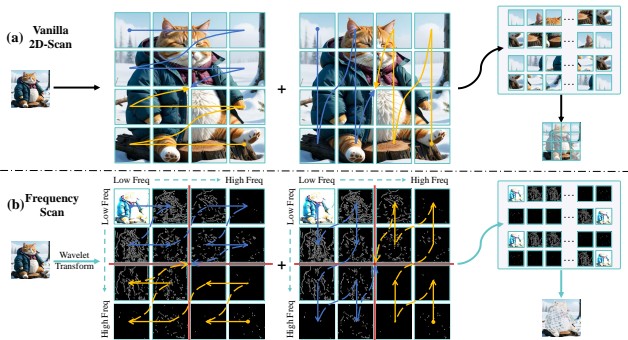

**Figure 3: The comparison between (a) the vanilla scanning strategy employed by VMamba [26] and (b) our frequency dimension strategy. Utilizing k-level wavelet packet transform, we decompose the input into $4^k$ ($k$=2 in the figure) frequency bands. It is then scanned along the frequency dimension in the spatial domain. This strategy introduces a new dimension to 2D-Mamba, allowing it to capture complex image details at different frequencies. It is noteworthy that we use a local scanning strategy similar to LocalMamba[18], which is more consistent with the form of WPT and allows strictly frequency-ordered scanning.**

This process converts spatial features into a complex component, illustrating the image's frequency components. The frequency domain representation $\mathcal{F}(x)$ can be decomposed into real $\mathcal{R}(x)$ and imaginary $\mathcal{I}(x)$ parts, leading to amplitude spectrum and phase spectrum, describing the image's frequency content:

$$\mathcal{A}(x)(u,v) = \sqrt{\mathcal{R}^2(x)(u,v) + \mathcal{I}^2(x)(u,v)},$$
$$\mathcal{P}(x)(u,v) = \arctan\left[\frac{\mathcal{I}(x)(u,v)}{\mathcal{R}(x)(u,v)}\right]. \quad (2)$$

Another tool, the Discrete Wavelet Transform (DWT), decomposes an image $I \in \mathcal{R}^{H \times W \times C}$ into four sub-bands representing low-frequency approximations and high-frequency details:

$$I_{LL}, I_{LH}, I_{HL}, I_{HH} = DWT(I). \quad (3)$$

Each sub-band, corresponding to different directional details, reduces the original image's dimensions by half. In contrast, the Wavelet Packet Transform offers a more detailed frequency content analysis by further decomposing all sub-bands at each level.

*3.2.2 State Space Models.* State Space Models (SSMs) are fundamental in translating one-dimensional inputs into outputs through latent states, utilizing a framework of linear ordinary differential equations. For a system with input $x(t)$ and output $y(t)$, the model dynamics are described by:

$$h'(t) = \mathbf{A}h(t) + \mathbf{B}x(t),$$
$$y(t) = \mathbf{C}h(t) + \mathbf{D}x(t), \quad (4)$$

where $\mathbf{A}, \mathbf{B}, \mathbf{C}$, and $\mathbf{D}$ are the model parameters. The discrete versions of these models, such as Mamba, incorporate a discretization step through the zero-order hold (ZOH) method while allowing the

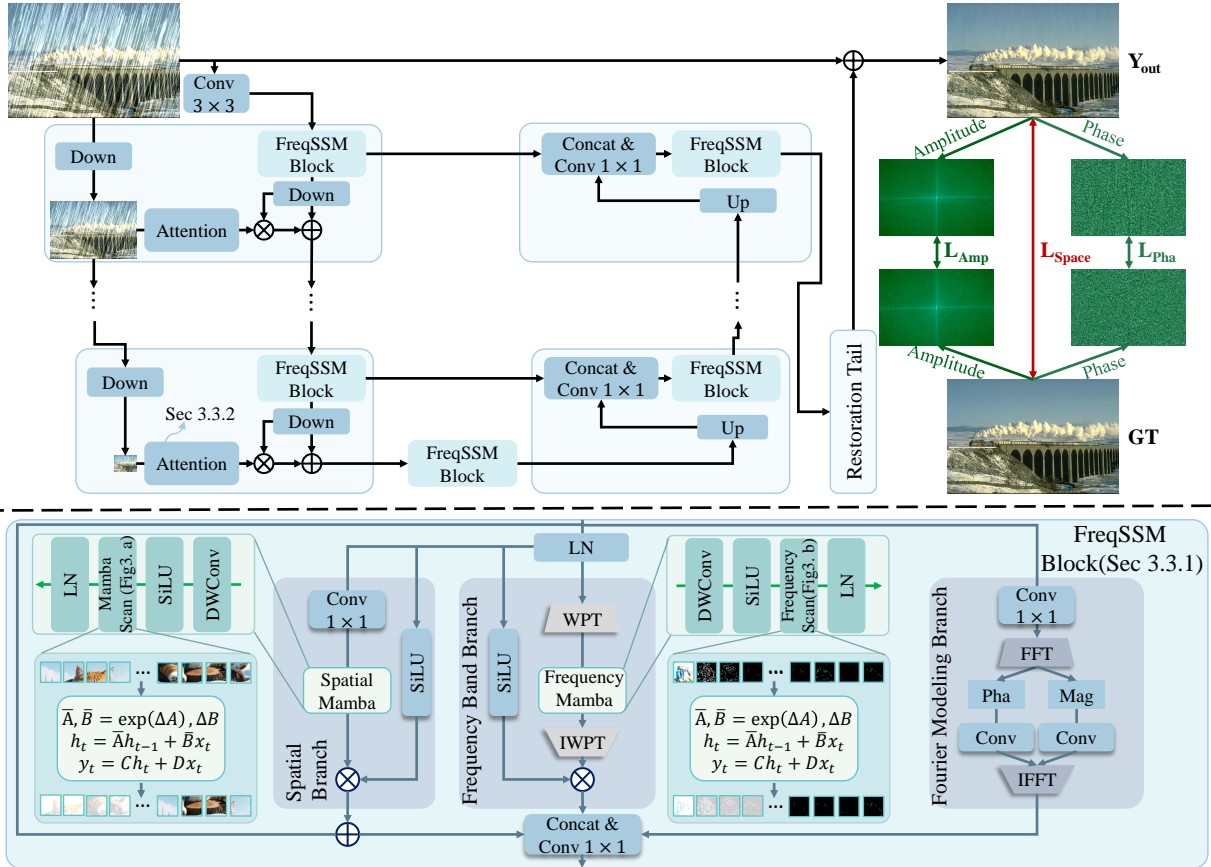

**Figure 4: The detailed architecture of our FreqMamba. The three-branch FreqSSM forms the basic block of the u-net architecture for global and local modeling. Multi-scale degradation priors are introduced into the training process at the encoder stage.**

models to scan and adjust to the input data adaptively through a selective scanning mechanism. This adaptability is particularly useful in complex applications like image restoration, where understanding contextual relationships between different image regions is crucial.

## 3.3 Architecture

We show our proposed method in Fig.4, which uses a multi-scale U-Net architecture with three-branch Frequency-SSM Blocks as the core components. At the same time, the input-dependent nature of Mamba is used to generate attention maps using degradation images of different scales, allowing the model to capture degradation distributions at different scales. Now we introduce the details of the different components of the network respectively.

*3.3.1 Frequency-SSM Block.* The frequency-SSM block is designed to address complex challenges by leveraging the synergy of three distinct branches: the Fourier Modeling Branch, the Spatial Branch, and the Frequency Band Branch.

**Fourier Modeling Branch.** For the Fourier Modeling Branch, the input feature $F_{in}$ first undergoes a convolution layer to produce $F_0$. $F_0$ is transformed into its Fourier spectrum via the Fast Fourier

Transform (FFT), which is then decomposed into amplitude spectrum $\mathcal{A}(F_0)$ and phase spectrum $\mathcal{P}(F_0)$. The amplitude and phase spectrum are refined through convolution blocks in the frequency domain and finally returned to the spatial domain through iFFT:

$$
\begin{aligned}
F_P &= ConvBlock(W_1(\mathcal{A}(F_0))), \\
F_A &= ConvBlock(W_1(\mathcal{P}(F_0))), \\
F_f &= \mathcal{F}^{-1}(F_P, F_A),
\end{aligned}
\tag{5}
$$

where $W_1(\cdot)$ donates a $1 \times 1$ convolution operation and $ConvBlock(\cdot)$ signifies a series of convolution operations and activation functions. This branch captures and processes the frequency domain representation of features, mastering the global recovery of images.

**Spatial Branch.** We first use Layernorm to process the input features $F_{in}$ to get $F_{LN}$. The features then pass through two parallel sub-branches. The first sub-branch simply SiLU activates them. The other sub-branch performs spatial Mamba on features after $1 \times 1$ convolution. Spatial Mamba consists of sequence : $DWConv \rightarrow SiLU \rightarrow Mamba\text{-}scan \rightarrow LN$, where Mamba-scan means vanilla 2D Mamba-scan shown in Fig. 3 (a). The outputs of two sub-branches are then element-wise multiplied to yield the spatial output $F_s$:

$$
F_s = MambaScan(W_1(F_{LN})) \odot SiLU(F_{LN}),
\tag{6}
$$

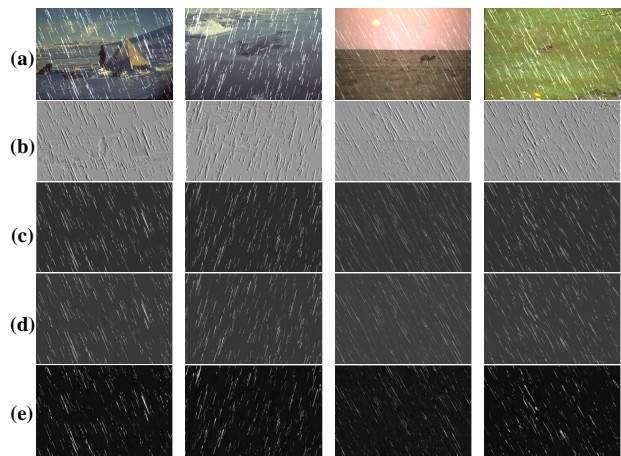

Figure 5: Visualization of degradation prior attention maps and three-branch features. (a) Rainy images. (b) Attention maps. (c), (d), and (e) are the feature maps of the spatial, frequency band, and Fourier modeling branch. The spatial branch (c) comprehensively recognizes raindrops but the boundaries are blurry. The Fourier branch (e) outputs high-contrast features and focuses more on larger rain streaks. Overall the frequency band branch (d) is somewhere in between.

where $\odot$ donates element-wise multiplication.

**Frequency Band Branch.** $F_{LN}$ is subjected to a 2-layer wavelet packet transform to capture the information of different frequency bands with the size of $\frac{H}{4}, \frac{W}{4}$. They are arranged back to the original resolution from top left to bottom right, then processed by frequency Mamba, which is unique in its frequency scan as shown in Fig. 3 (b). Since the strategy in Fig. 3 (a) cannot completely match WPT along the low to high frequency, we use a strategy similar to LocalMamba [18]. We divide the wavelet features into four blocks (red line in Fig. 3 (b)) and scan separately block by block.

The output is returned to the spatial domain after the corresponding k-layer wavelet packet inverse transformation and lastly multiplied element by element with $SiLU(F_{LN})$ to obtain the frequency band scanning output $F_b$. This procedure is encapsulated as:

$$F_b = IDWT(FreqScan(DWT(F_{LN}))) \odot SiLU(F_{LN}), \quad (7)$$

where $WPT$ and $IWPT$ donate discrete wavelet packet transform and inverse discrete wavelet packet transformation, respectively. We regard the frequency dimension of Mamba modeling as a transition between Mamba and Fourier modeling to achieve global degradation processing and local detail restoration seamlessly.

$F_{in}$ is added with the output of the spatial branch using a residual connection. By concatenating the output features of three branches and applying a final $1 \times 1$ convolution operation, the block achieves a harmonized synthesis of the features. In Fig. 5, we present visualizations of the features from different branches to illustrate the variations in rain capture across each branch.

### 3.3.2 Data-dependent Degradation Prior Attention Map.
In various regions of the degraded image, the challenge of recovery varies significantly, influenced by the degradation distribution and

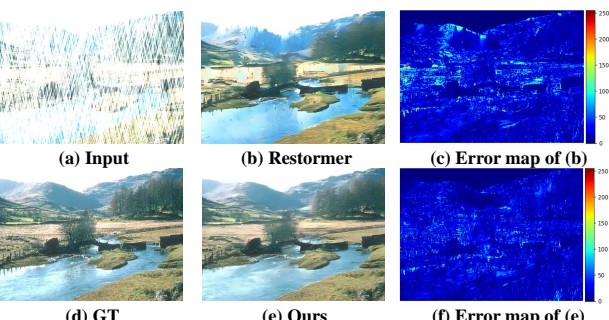

Figure 6: Error map (c) is the difference between the restored image (b) by Restormer[51] and the ground truth (d). There are significantly larger errors in mountainous areas with dense rain streaks and complex backgrounds, reflecting the uneven recovery difficulty and, more importantly, the need to model degradation priors. Error map (f) is the difference between our restoration result and GT with fewer huge error values.

the image background's complexity. As illustrated in Fig. 6, the restoration performed by Restormer [51] exhibits a huge error with GT in areas such as mountains, characterized by dense rain patterns and intricate backgrounds. This discrepancy stems from the absence of explicit modeling of the degradation in the rainy images.

To address this issue, we leverage the unique data-dependent property of the Mamba module, which allows it to dynamically focus on or disregard specific input features based on their significance. We introduce an innovative approach wherein we utilize the Mamba module to generate degradation priors, thereby enhancing our model's ability to discern and address the varying degrees of degradation across different regions of the image. This process involves the generation of degradation priors through spatial 2D scanning of input images at multiple scales. These degradation priors are then element-wise multiplied with features corresponding to the same scale and subsequently summed with the features:

$$\begin{aligned} M_{atten} &= Mamba(W_1(I_{in})), \\ F &= F \odot M_{atten} + F, \end{aligned} \quad (8)$$

where $I_{in}$ donates low-resolution rainy image and $F$ donates feature at the same scale. By incorporating input features of different scales, our model can effectively delineate the possible locations of degradation at various levels of granularity. As shown in Fig. 5, we show some rainy images and the corresponding degradation attention maps. The rain streaks are highlighted in the attention maps.

In essence, this approach harnesses the inherent adaptability of the Mamba module to dynamically allocate attention to different regions of the input image, thereby enabling our model to better address the intricate of degradation distribution across the image.

### 3.4 Loss Function

In addition to new modules and degradation priors, we also introduce new loss functions to optimize the training process of the network to achieve good results in both spatial and frequency domains. The

**Table 1: Quantitative comparison (PSNR/SSIM) for Image Deraining on five benchmark datasets. The highest and second-highest performances are marked in bold and underlined. '-' indicates the result is not available.**

| Method | Venue | Rain100H [47] | | Rain100L [47] | | Test2800 [7] | | Test1200 [53] | | Param(M) | GFlops |
|---|---|---|---|---|---|---|---|---|---|---|---|
| | | PNSR ↑ | SSIM ↑ | PNSR ↑ | SSIM ↑ | PNSR ↑ | SSIM ↑ | PNSR ↑ | SSIM ↑ | | |
| DerainNet [6] | TIP'17 | 14.92 | 0.592 | 27.03 | 0.884 | 24.31 | 0.861 | 23.38 | 0.835 | 0.058 | 1.453 |
| UMRL [49] | CVPR'19 | 26.01 | 0.832 | 29.18 | 0.923 | 29.97 | 0.905 | 30.55 | 0.910 | 0.98 | - |
| RESCAN [24] | ECCV'18 | 26.36 | 0.786 | 29.80 | 0.881 | 31.29 | 0.904 | 30.51 | 0.882 | 1.04 | 20.361 |
| PreNet [35] | CVPR'19 | 26.77 | 0.858 | 32.44 | 0.950 | 31.75 | 0.916 | 31.36 | 0.911 | 0.17 | 73.021 |
| MSPFN [19] | CVPR'20 | 28.66 | 0.860 | 32.40 | 0.933 | 32.82 | 0.930 | 32.39 | 0.916 | 13.22 | 604.70 |
| SPAIR [33] | ICCV'21 | 30.95 | 0.892 | 36.93 | 0.969 | 33.34 | 0.936 | 33.04 | 0.922 | - | - |
| MPRNet [52] | CVPR'21 | 30.41 | 0.890 | 36.40 | 0.965 | 33.64 | 0.938 | 32.91 | 0.916 | 3.64 | 141.28 |
| Restormer [51] | CVPR'22 | 31.46 | 0.904 | 38.99 | 0.978 | 34.18 | 0.944 | 33.19 | 0.926 | 24.53 | 174.7 |
| Fourmer [56] | ICML'23 | 30.76 | 0.896 | 37.47 | 0.970 | - | - | 33.05 | 0.921 | 0.4 | 16.753 |
| IR-SDE [28] | ICML'23 | 31.65 | 0.904 | 38.30 | 0.980 | 30.42 | 0.891 | - | - | 135.3 | 119.1 |
| MambaIR [13] | arxiv'24 | 30.62 | 0.893 | 38.78 | 0.977 | 33.58 | 0.927 | 32.56 | 0.923 | 31.51 | 80.64 |
| VMambaIR [36] | arxiv'24 | 31.66 | 0.909 | 39.09 | 0.979 | 34.01 | 0.944 | 33.33 | 0.926 | - | - |
| FreqMamba (Ours) | - | **31.74** | **0.912** | **39.18** | **0.981** | **34.25** | **0.951** | **33.36** | **0.931** | 14.52 | 36.49 |

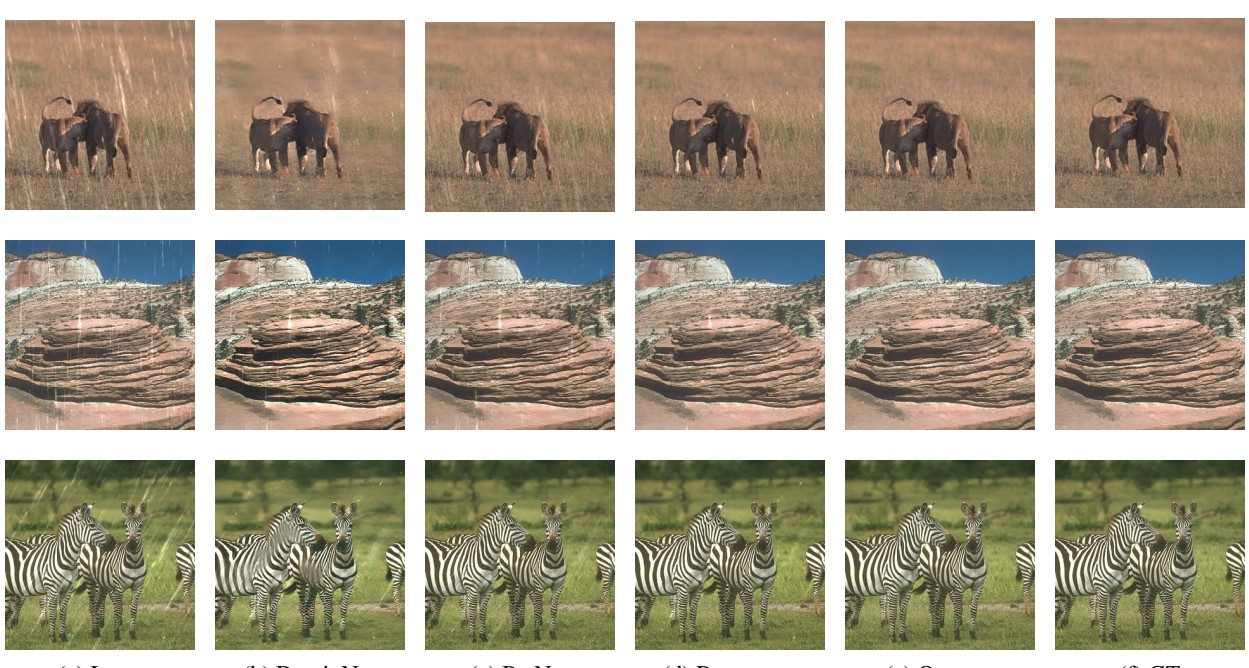

(a) Input     (b) DerainNet     (c) PreNet     (d) Restormer     (e) Ours     (f) GT

**Figure 7: Visual quality comparison on an image from Rain100L[47]. Zoom in for better visualization.**

loss function consists of three parts: a spatial domain loss, a phase spectrum loss, and an amplitude spectrum loss.

In the spatial domain, we use L1 loss between the final output and GT to achieve supervision. At the same time, to achieve better global information reconstruction, we calculate the L1 loss on the amplitude spectrum and phase spectrum respectively to obtain the amplitude and phase spectrum loss, expressed as:

$$
\begin{aligned}
L_{spa} &= \|Y_{out} - X_{gt}\|_1, \\
L_{amp} &= \|\mathcal{A}(Y_{out}) - \mathcal{A}(X_{gt})\|_1, \\
L_{pha} &= \|\mathcal{P}(Y_{out}) - \mathcal{P}(X_{gt})\|_1,
\end{aligned}
\tag{9}
$$

Finally, the overall composition of the loss function can be briefly expressed as follows:

$$
L_{total} = L_{spa} + \alpha L_{amp} + \beta L_{pha},
\tag{10}
$$

where $\alpha$ and $\beta$ are both empirically set to 0.05 in our implementation.

## 4 EXPERIMENTS

In this section, we evaluate our method through extensive experiments. First, We describe the experimental setup. Then we present the comparison results between our method and state-of-the-art approaches qualitatively and quantitatively. Lastly, a detailed ablation analysis of our proposed method is conducted.

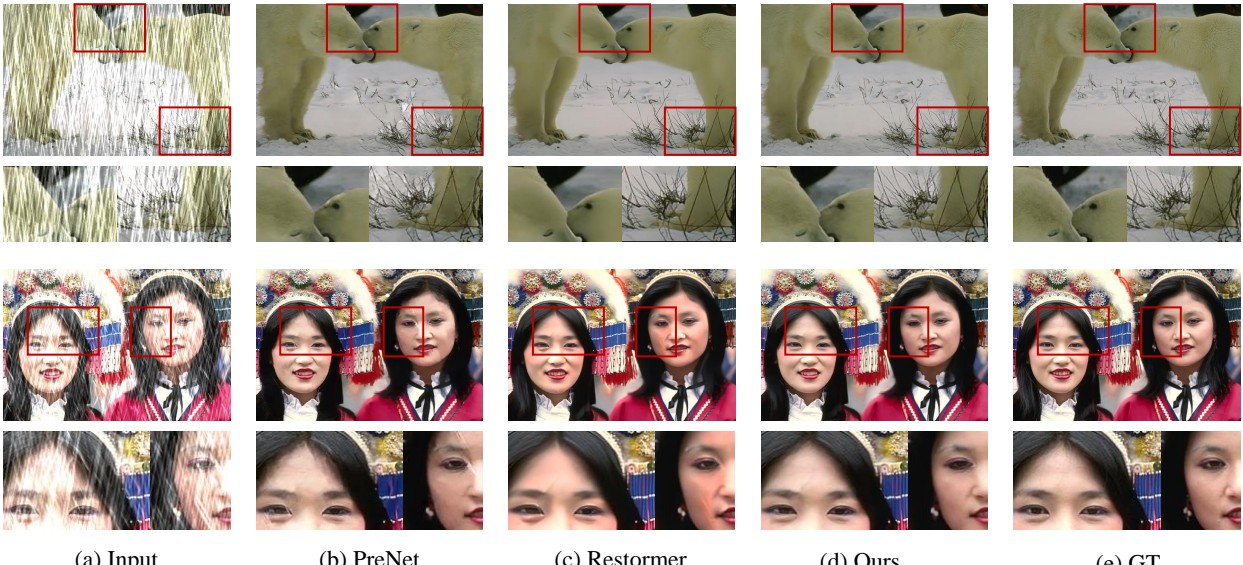

|        (a) Input        |        (b) PreNet        |        (c) Restormer        |        (d) Ours        |        (e) GT        |

**Figure 8: Visual quality comparison on an image from Rain100H[47]. Zoom in for better visualization.**

## 4.1 Datasets and Implementation

**Datasets.** For validation, we train and validate our model on the widely-used datasets Rain100H[47], Rain100L[47], Test1200[53], and Test2800[7] datasets. Rain100L is selected from BSD200 [30] with only one type of rain streaks, which consists of 200 image pairs for training and 100 image pairs for testing. Rain100H contains 1800 image pairs for training and 100 image pairs for testing with five types of streak directions. test1200[53] has three groups of 12,000 light, medium, and heavy rainy images for training, with each group 4,000 images. The same three groups of 1,200 images are used for testing. Test2800[7] contains 14000 image pairs, of which 12,600 pairs are used for training and 1,400 for testing.

**Implementation details.** Our model was implemented using the Pytorch framework and executed on NVIDIA RTX 3090 GPUs. The number of blocks in each layer affects the amount of model parameters and rain removal performance. After weighing the balance, we set blocks of each layer to [2, 3, 3, 4, 3, 3, 2], which can achieve good performance with a reasonable amount of parameters. We use the progressive training strategy. Taking the RAIN100L dataset as an example, we set the total number of iterations to 75000 and set the image size to [160, 256, 320, 384] while the corresponding batch sizes were [8, 4, 2, 1]. We use the Adam[21] optimizer with default parameters to train the network by minimizing the loss function $L_{total}$. The initial learning rate is set to $3 \times e{-}4$ and then gradually decays to $1 \times e{-}6$ using the cosine annealing strategy.

## 4.2 Comparison with State-of-the-art Methods

In this section, we compare our method with state-of-the-art deraining approaches: DerainNet [6], UMRL [49], RESCAN [24], PreNet [35], MSPFN [19] , SPAIR [33] , MPRNet [52] , Restormer[51], Fourmer [56], IR-SDE [28], MambaIR [13], and VMambaIR [36] .

**Quantitative Comparison.** Following [51], we compute Peak Signal-to-Noise Ratio(PSNR) and Structural Similarity(SSIM) scores using

**Table 2: Ablation study for investigating the components of FreqMamba. The former three columns represent the three branches. Map refers to the attention map.**

| Fourier | Frequency Band | Spatial | Map | PSNR | SSIM |
|---------|---------------|---------|-----|-------|--------|
|         | ✓             | ✓       | ✓   | 38.85 | 0.9782 |
| ✓       |               | ✓       | ✓   | 39.08 | 0.9801 |
| ✓       | ✓             |         | ✓   | 37.33 | 0.9758 |
| ✓       | ✓             | ✓       |     | 39.11 | 0.9803 |
| ✓       | ✓             | ✓       | ✓   | 39.18 | 0.9814 |

the Y channel in YCbCr color space. Tab. 1 reports the performance evaluation on the four datasets. As can be seen, our method achieves the best performance among all the baseline algorithms.

**Qualitative Comparison.** To demonstrate the enhanced fidelity and level of detail exhibited by images generated by our proposed FreqMamba model for the image rain removal task, we compare the visual quality of challenging degraded images from the Rain100L in Fig. 7 and Rain100H datasets in Fig. 8. When faced with complex or very severe rain streaks, our method achieves almost perfect results. Compared with previous methods, our FreqMamba achieves the perfect performance of global and local recovery. For example, by zooming in the red box region in Fig. 8, our method removes more rain streak residue while better restoring texture details. We provide more visualization results in the supplementary material.

## 4.3 Ablation Studies

We performed ablations on different components of the model on the Rain100L dataset. More results are provided in the supplementary.

**Investigation of space mamba branch.** To verify the efficacy of the space mamba branch, we replace Mamba blocks with standard convolution layers and keep other parts unchanged. The result presented in Tab.2 shows that the performance has dropped significantly, as

**Table 3: Ablation study of the contribution of loss functions.**

| Loss function | PSNR ↑ | SSIM ↑ |
|---|---|---|
| w/o $L_{freq}$ | 39.09 | 0.9795 |
| w/o $L_{amp}$ | 39.12 | 0.9809 |
| w/o $L_{pha}$ | 39.12 | 0.9807 |
| Full Loss | 39.18 | 0.9814 |

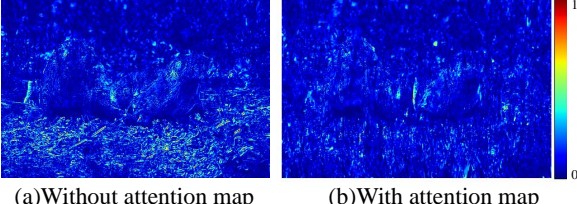

    (a)Without attention map      (b)With attention map

**Figure 9: The visualization of the errors between the restored image and corresponding GT. With the employment of the degradation prior attention map, the errors are greatly reduced.**

the limited receptive field of standard convolution results in its poor modeling capability.

**Investigation of Fourier branch.** We remove the Fourier branch, leaving the spatial and frequency band mamba branch. The lack of global overall modeling ability reduces model performance by 0.33 dB, which shows the importance of modeling global information in the Fourier domain. It's consistent with our original intention.

**Investigation of frequency band branch.** We remove the frequency branch, leaving the spatial mamba and Fourier branch. The model performance has dropped by 0.1 dB, which shows the importance of the new frequency dimension.

**Investigation of attention map.** Our degradation prior attention strategy adaptively learns degradation distributions based on rainy images at different scales. To verify its effectiveness, we conduct ablation experiments in Tab. 2. It can be clearly seen that when equipped with the attention map, the algorithm gives better results than that without using degradation prior. Further, we show an error map in Fig. 9 with or without the attention map, which demonstrates the effectiveness of our degradation prior attention strategy. More comparisons can be found in the supplementary.

**Investigation of loss function.** The frequency loss aims to directly emphasize global frequency information optimization. We remove it and its two components in Tab. 3 respectively to check the validity. The results show that removing it decreases performance metrics, indicating its importance.

## 4.4 Extensions on Other Tasks

To demonstrate the potential of our FreqMamba, we extend it to low-light image enhancement and Real-world image dehazing.

    **Extension on low-light image enhancement.** Low-light image enhancement mainly focuses on illuminating the darkness of the scene and removing amplified noise. Our three-branch structure copes with this scenario well. We adopt LOL-V1 and LOL-V2-Synthetic datasets to evaluate the performance of our method. Several low-light image enhancement methods are selected for comparison: RetinexNet [41], KinD [55], ZeroDCE [12], KinD ++ [54],

**Table 4: Quantitative results of different methods on the LOL-V1 and LOL-V2-Syn dataset.**

| Method | LOL-V1 [47] | | LOL-V2-Syn [47] | |
|---|---|---|---|---|
| | PNSR ↑ | SSIM ↑ | PNSR ↑ | SSIM ↑ |
| RetinexNet [41] | 18.38 | 0.7756 | 19.92 | 0.8847 |
| KinD [55] | 20.38 | 0.8248 | 22.62 | 0.9041 |
| ZeroDCE [12] | 16.80 | 0.5573 | 17.53 | 0.6072 |
| KinD++ [54] | 21.30 | 0.8226 | 21.17 | 0.8814 |
| URetinex-Net[43] | 21.33 | 0.8348 | 22.89 | 0.8950 |
| FECNet [17] | 22.24 | 0.8372 | 22.57 | 0.8938 |
| SNR-Aware [46] | 23.38 | 0.8441 | 24.12 | 0.9222 |
| FreqMamba (Ours) | **23.57** | **0.8453** | **24.46** | **0.9355** |

URetinex-Net [43], FECNet [17], and SNR-Aware [46]. Tab. 4 shows the quantitative comparison.

    **Extension on real-world image dehazing.** Real-world dehazing aims to restore a clean scene from a real-world hazy image. For this task, we apply two datasets: Dense-Haze [1]and NH-HAZE [2]. We compare our method with others including DCP [15], DehazeNet [4], GridNet[25], MSBDN [5], and AECR-Net [42]. We present the quantitative comparison in Tab. 5.

**Table 5: Quantitative results of different methods on the Dense-Haze and NH-HAZE dataset.**

| Method | Dense-Haze [1] | | NH-HAZE [2] | |
|---|---|---|---|---|
| | PNSR ↑ | SSIM ↑ | PNSR ↑ | SSIM ↑ |
| DCP [15] | 10.06 | 0.3856 | 10.57 | 0.5196 |
| DehazeNet [4] | 13.84 | 0.4252 | 16.62 | 0.5238 |
| GridNet [25] | 13.31 | 0.3681 | 13.80 | 0.5370 |
| MSBDN [5] | 15.37 | 0.4858 | 19.23 | 0.7056 |
| AECR-Net [42] | 15.80 | 0.4660 | 19.88 | 0.7173 |
| FreqMamba (Ours) | **17.35** | **0.5827** | **19.93** | **0.7372** |

## 5 CONCLUSION

In this work, we introduce FreqMamba, an innovative deraining network that seamlessly integrates spatial domain sequence modeling and frequency domain global modeling to address the challenge of image deraining. The core of the FreqMamba network utilizes the unique frequency SSM block. Meanwhile we leverage Mamba's input dependency properties to generate attention maps across multiple scales. This integration enables a nuanced understanding and processing of rain streaks, distinguishing our approach from existing methods. Our comprehensive experiments on various datasets highlight the effectiveness and efficiency of FreqMamba. Notably, the model is not only good at removing rain from images but also preserves the integrity and detail of the underlying scene. In applications where final image quality is critical, the balance of performance and fidelity is critical. Furthermore, FreqMamba has applications beyond rain removal, demonstrating its versatility in a variety of image restoration tasks. This adaptability illustrates the robustness of the underlying architecture and its potential as a foundational model for future image restoration research.

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
