# OpenReview forum: "FreqMamba: Viewing Mamba from a Frequency Perspective for Image Deraining"
_acmmm.org/ACMMM/2024/Conference — MM2024 Poster_

### Official Review · Reviewer_6ETt · 2024-05-16

**Rating:** 3
**Confidence:** 3

**Summary:**

This paper proposes to utilize the latest Mamba architecture together with Fourier transform to address image deraining task. It introduces a Frequency-based Mamba framework, which consist of three branches, i.e., spatial mamba, frequency band mamba and fourier modeling. Extensive experiments are conducted on diverse low-level vision tasks, e.g., image deraining, image dehazing, low-light image enhancement.

**Strengths:**

It is an interesting and pioneer work to introduce SSMs into low-level vision tasks.
Extensive experiments on image deraining, dehazing, low-light image enhancement are conducted, demonstrating its effectiveness.
Many visualization results of frequency and attention maps are displayed to better demonstrate the effectiveness of the proposed modules.

**Limitations:**

1. The motivation of using Mamba is unclear. Mamba is designed to causally process text data in NLP tasks. However, for non-causal data 2D images, Mamba takes no advantage and may destroy the spatial relations by flattening them into sequences. Moreover, Mamba have superior performance when coming to long sequences. For 2D images without high resolution, self-attention is sufficient to address such short sequences in image-deraining task. Please further clarify.
2. Related work on State Space Models is insufficient. Many relevant works have sprung out, such as MambaIR, VMambaIR, T-Mamba, SegMamba, Vivim. The author should clarify the core differences compared to other works.
3. Sec. 3.1 Motivation is cumbersome to understand. The definition of global degradation is unclear. The author should explain more specifically. In addition, the author stated that frequency domain operations only focus on the global dependency while Mamba can complement it. However, as far as I know, Mamba is introduced to replace self-attention for efficient global modeling in many works. There is no analysis or experimental results to validate the effectiveness of Mamba in modeling local dependencies.
4. In Fig. 7&8, the author should also display the results of the second-best method VMambaIR.
5. In Tab. 2, removing the spatial branch brings a significant drop in PSNR, SSIM while Fourier and FrequencyBand contribute little gain. It seems that vanilla Mamba is taking the heavy lifting.

**Suitability:**

3

---

### Official Review · Reviewer_G9GE · 2024-05-19

**Rating:** 5
**Confidence:** 3

**Summary:**

The paper proposes a novel method for image deraining, called FreqMamba, which leverages frequency analysis and the Mamba model to achieve state-of-the-art results. The method consists of three branches: spatial Mamba, frequency band Mamba, and Fourier global modeling, which work together to capture both local and global degradation patterns. The authors conduct extensive experiments to validate the effectiveness of their approach, comparing it to several state-of-the-art methods on various datasets. The results show that FreqMamba outperforms the other methods both visually and quantitatively, achieving better spectral fidelity and fewer residues.

**Strengths:**

Integrating frequency analysis with spatial modeling via Mamba is a novel approach for image deraining. Exploiting the complementary strengths of frequency and spatial domains could lead to improved global and local degradation modeling.
The paper provides a sound theoretical foundation, describing frequency analysis concepts like Fourier transform, wavelet transform, and state space models clearly. The proposed three-branch architecture (Fourier Modeling, Spatial Mamba, Frequency Band Mamba) is well-motivated.

**Limitations:**

While the integration of different components is novel, the individual components (Fourier transform, wavelet transform, Mamba) are well-established techniques.
The proposed architecture appears to be computationally expensive due to the multiple branches and transformations involved, which could limit its practical applications.

**Suitability:**

2

---

### Official Review · Reviewer_SJTY · 2024-05-21

**Rating:** 3
**Confidence:** 3

**Summary:**

FreqMamba utilizes the complementarity between Mamba and frequency analysis to de-rain images, extends Mamba with frequency bands to take advantage of frequency correlation, and connects it with the Fourier transform to model global degradation. It is visually and quantitatively superior to the most advanced methods (e.g. VMambaIR, MambaIR)

**Strengths:**

The paper proposes FreqMamba which lies in a three-branch (Spatial Mamba + Frequency Band Mamba + Fourier Modeling) structure. These branches form a robust architecture for tackling the challenge of image deraining..

**Limitations:**

1. The improvement of Frequency brand is marginal (shown in Table 2), which makes me concerned about the the novelty of the paper.
2. Why not use FFT in Frequency brand? DWT is still a local transform.
3. The paper lacks the reference for Frequency Domain Learning, e.g. FFC [1], LaMa [2], DeepRFT [3] and so on.

[1] Fast fourier convolution. NeurIPS, 2020.

[2] Resolution-robust large mask inpainting with fourier convolutions. WACV, 2022.

[3] Intriguing findings of frequency selection for image deblurring. AAAI, 2023.

**Suitability:**

2

---

### Official Review · Reviewer_n6bh · 2024-05-24

**Rating:** 4
**Confidence:** 4

**Summary:**

This paper proposes FreqMamba, a method that leverages the complementary between Mamba and frequency analysis for image deraining.  Specifically, FreqMamba comprises three branches: Spatial Mamba, Frequency Band Mamba, and Fourier Modeling. Spatial Mamba focuses on extracting details and correlations from original image features. Frequency Band Mamba dissects input features into various frequency bands using wavelet packet transform. Fourier Modeling utilizes the Fourier Transform for global analysis capabilities and refining overarching degradation patterns.

**Strengths:**

**Experimental validation**

- This paper presents state-of-the-art results on the Rain100L, Rain100H, Test2800 and Test1200 benchmark datasets for single image deraining. In addition, the results in low light enhancement and removal of non-uniform haze were also additionally demonstrated.
- This paper provides a rich analysis of visual results.

**Technical correctness**
- The paper formulation is technically sound.

**method novelty**
- This paper combines Mamba from the perspective of frequency domain to explore the characteristics and connections of degraded images in the frequency domain state space. It has certain inspiration and reference value in low level visual tasks.

**Limitations:**

- Although this paper compares many methods on the derivation task, there is still a lack of comparison with some representative methods, such as DRSformer [1], IDT [2]. In addition, there is a lack of comparison with Retinexformer [3] in low light image enhancement, and a lack of comparison with the last two years in the NH-HAZE dataset.

- This paper for image deraining whether experiment within the following benchmark data: DID Data, DDN Data and SPA Data?

- Is the comparison method in this article retrained while maintaining the same fair participation?

- This paper lacks generalization experiments on real-world datasets.

[1] Chen X, Li H, Li M, et al. Learning a sparse transformer network for effective image deraining[C]//Proceedings of the IEEE/CVF Conference on Computer Vision and Pattern Recognition. 2023: 5896-5905.

[2] Xiao J, Fu X, Liu A, et al. Image de-raining transformer[J]. IEEE Transactions on Pattern Analysis and Machine Intelligence, 2022.

[3] Cai Y, Bian H, Lin J, et al. Retinexformer: One-stage retinex-based transformer for low-light image enhancement[C]//Proceedings of the IEEE/CVF International Conference on Computer Vision. 2023: 12504-12513.

**Suitability:**

3

---

### Meta-Review · Area_Chair_gYZL · 2024-07-02

**Recommendation:** Accept (Poster)
**Confidence:** 5

**Metareview:**

The overall quality of the paper is high. It presents a well-structured approach combining Mamba and frequency domain analysis, leading to impressive results on standard benchmarks. The experimental validation is robust, and the theoretical foundation is sound.

The paper is generally well-written, but some sections could benefit from improved clarity and additional explanations.

The paper presents a novel integration of Mamba with frequency analysis for image deraining, which is original and contributes to the field.

The paper presents a novel and effective approach to image deraining, with strong experimental results. However, addressing the mentioned concerns will strengthen the paper and provide a more comprehensive evaluation of its contributions.